# Data-Driven Medicine in the Diagnosis and Treatment of Infertility

**DOI:** 10.3390/jcm11216426

**Published:** 2022-10-29

**Authors:** Ines de Santiago, Lukasz Polanski

**Affiliations:** 1Cancer Research UK, Cambridge Institute, University of Cambridge, Cambridge CB2 3EA, UK; 2Department of Obstetrics and Gynaecology, North West Anglia Foundation Trust, Peterborough PE3 9GZ, UK

**Keywords:** infertility, machine learning, big data, P4 medicine

## Abstract

Infertility, although not a life-threatening condition, affects around 15% of couples trying for a pregnancy. The increasing availability of large datasets from various sources, together with advances in machine learning (ML) and artificial intelligence (AI), are enabling a transformational change in infertility care. However, real-world applications of data-driven medicine in infertility care are still relatively limited. At present, very little can prevent infertility from arising; more work is required to learn about ways to improve natural conception and the detection and diagnosis of infertility, improve assisted reproduction treatments (ART) and ultimately develop useful clinical-decision support systems to assure the successful outcome of either fertility preservation or infertility treatment. In this opinion article, we discuss recent influential work on the application of big data and AI in the prevention, diagnosis and treatment of infertility. We evaluate the challenges of the sector and present an interpretation of the different innovation forces that are driving the emergence of a systems approach to infertility care. Efforts including the integration of multi-omics information, collection of well-curated biological samples in specialised biobanks, and stimulation of the active participation of patients are considered. In the era of Big Data and AI, there is now an exciting opportunity to leverage the progress in genomics and digital technologies and develop more sophisticated approaches to diagnose and treat infertility disorders.

## 1. Introduction

Infertility, defined as failure to conceive after 1 year or more of regular unprotected sexual intercourse, is estimated to affect 15% of couples of reproductive age [1,2], with regional variations mainly affected by socio-economic status [3]. Although robust epidemiological data are lacking, there is a possibility that this trend will worsen, especially in sub-Saharan Africa and South Asia [4]. In most cases of infertility, the cause is known, with tubal and male factors being most common. The tubal factor is more common in areas with a high incidence of sexually transmitted infections and is strongly correlated with intra-abdominal infections leading to adhesion formation. In recent decades, semen parameters have been deteriorating, possibly contributing to the rise in male factor infertility [5]. In approximately a quarter of infertility cases, the cause cannot be established using conventional diagnostic methods [6]. The main cause for the increase in infertility rates in recent decades, however, is the postponement of childbearing due to various socio-economic and personal reasons [7]. Based on The Organization for Economic Co-operation and Developments (OECD) statistics, the mean maternal age of first child birth has increased by from 2 to 5 years from 1970 to 2017, with majority of women now having their first child at the age of 30 or more [8]. When the presumed cause of subfertility is established, or after a certain period of unexplained infertility, the only treatment of choice is ART in the form of in vitro fertilization (IVF) or intracytoplasmic sperm injection (ICSI). The failure of less-invasive treatment options such as intrauterine insemination (IUI) or ovulation induction (OI), would also lead to the above therapies. Though ART has come a long way since the first IVF baby was born in 1978, the overall success rates per embryo transfer are in the region of 35% [9]. This is marginally better than nature, as the estimated monthly fecundity rate is 20% per cycle [10].

In recent decades, we have witnessed massive technological changes and innovations that have influenced the field of infertility, particularly the ways we perform IVF, handling and evaluating the oocytes, the sperm and the embryos. Examples of such advancements include the introduction of ICSI technology for male factor infertility in 1991 [11] and the introduction of pre-implantation genetic testing or screening (PGT/S) of embryos in the 1990s [12]. Machine learning (ML), artificial intelligence (AI), and other modern statistical data-mining methodologies are now providing new opportunities in precision medicine to improve infertility care [13,14]. The term of ‘P4 medicine’ subsumes the preventative, personalised, participatory and predictive aspects of medicine, and was initially promulgated 17 years ago by the systems biologist Lee Hood [15]. This concept has mainly been applied to cancer and chronic diseases such as cardiovascular diseases, which represent the major causes of death globally. The increasing availability of advanced technologies allowing for screening of the genome, transcriptome, microbiome and metabolome should find their way into infertility investigations and treatment. The utilization of big data and ML to screen the results of molecular phenotyping and other clinical analyses, as well as population-based data and registries, will allow for us to continue developing insights into additional causes of infertility and establish improved and personalised treatments. Understanding why infertility occurs may reveal some unknown factors, which could be used to make diagnoses and treatments better, faster, less expensive and, as such, grant infertile couples the family they so desire.

In this paper we refer to this notion as a ‘systems medicine approach’ to infertility (Figure 1). This approach can be defined as the use of advanced computer algorithms such as ML/AI and big-data-mining techniques to integrate the diverse factors (genetic, lifestyle, environmental) implicated in this disease, with the aim of developing better tools for infertility prevention, prediction, diagnostics and treatment.

We aim to describe the key developments, promises and limitations of this approach, and identify possible avenues to its implementation, with a detailed discussion of four innovation forces that are driving the emergence of a system approach to the infertility sector:A conceptual shift from a disease-centric to a health-centric model: How infertility care is moving beyond a disease-based reactive model to a pro-active model focused on enhancing patients’ health and well-being.Better prevention, diagnosis and treatments: Sophisticated big data analysis of cohorts have allowed for the development of better strategies for diagnosis and treatments.ML/AI in ART Treatments: How ML/AI are currently being used to improve IVF across almost all stages of the treatment process.The participatory citizen: How the individual is empowered to drive their own reproductive health and well-being [16].

## 2. A Conceptual Shift from a Disease-Centric to a Health-Centric Model

The chance of natural conception for a couple with normal fertility, as a result of intercourse without contraception during the fertile phase, is about 20–25% for any given month [17]. Perhaps due to an emphasis on preventing unwanted pregnancies and sexually transmitted diseases, most people are well-informed about contraceptive precautions but are generally unaware of or unconcerned with their fertility health until they start trying to conceive. At present, the approach to infertility is to treat rather than prevent, with very little effort put into the latter. Tubal factor, one of the leading causes of infertility, is mainly caused by sexually transmitted infections. Prevention of these (use of condoms) and or early treatment reduces the insult, leading to tubal damage, which may cause tubal factor infertility in 15% of cases, following a single infection [18,19]. Another example of a simple intervention to improve chances of pregnancy is a modest decrease of even 5% of total body weight in cases of obesity associated with anovulatory polycystic ovarian syndrome (PCOS), which has been shown to restore ovulation in a large proportion of women, potentially sparing them the need for ART [20,21].

Access to accurate data and information about reproductive health can empower patients and practitioners with the necessary knowledge, thereby motivating behavioral changes and a shift from a disease-based model to a health-model of fertility. Some practical examples of this include the introduction of fertility check-ups by IVF clinics, allowing for patients to use a combination of blood tests and ultrasound scans to determine their infertility/fertility potential. In this context, markers of ovarian reserve (anti-Mullerian hormone (AMH) and antral follicle count (AFC)) are used to determine the expected duration of the woman’s or couples’ fertile years to guide them as to when to try and conceive in order to achieve the desired family size. When the ovarian reserve is found to be poor, fertility preservation may be advised. This can be carried out either in the form of oocyte or embryo freezing, with the outcomes of both methods being very promising.

Another example of the shift in mindset from a disease-based to a health-based model of infertility care is the increasing use of oocyte freezing. In 2018, in the UK alone, 1933 oocyte-freezing cycles were carried out, representing an almost quadrupling of numbers from 2013 [22]. In 74% of cases, these were privately funded, indicating that the likely reason for oocyte freezing was social circumstances [23], i.e., as a way of preventing problems that may arise in the future. However, it is often older women who seek egg-freezing, while it is in younger women (below the age of 35) that success is considered to be optimal [23]. Furthermore, irrespectively of how oocytes or embryos are cryopreserved for future use, there is no guarantee that the person will be able to have a biological child, and the chances will decrease with the women’s age. Consequently, data and information accessible to people at an early stage of their reproductive lives should be a driving force in the decision making process. For instance, it has been estimated that, when cryopreserving oocytes, a woman at the age of 34 would need to store 20 mature oocytes to have a 90% likelihood of having at least one child. For women of 42, this likelihood would be only 37% [24]. While it is possible for a 34-year-old woman to achieve 20 mature oocytes during one stimulation, it may be risky, and often requires two or more fresh controlled ovarian hyperstimulation (COH) cycles. In women of 42 years old, multiple cycles will be required to achieve this number. This, unfortunately, forms a significant financial barrier for most women, as the average cost of COH, freezing and storage of oocytes in the UK is estimated to be £4600 per cycle [25]. In addition to this, in the UK, the need to overcome legal limitations of the 10-year storage limit on gametes must be addressed for women to be able to store their eggs without the pressure of using them within the stated timeframe, which is mostly a consideration for women in their early or mid 20s. This matter has undergone public debate and will be addressed in the UK parliament soon. 

As the value of preventive, predictive and personalized medicine is increasingly being recognized, there is a great opportunity to encourage this paradigm change in infertility, i.e., away from a disease-centred vision to a preventative health-centred vision. From this perspective, the role of data and information is key to allow for patients to take informed actions sooner. This essential participatory aspect of the patient is discussed below. A big limitation of the preventative aspect of infertility is related to the fact that it is assisted and, as such, artificial. This may form barriers in uptake due to religious or personal beliefs and the perceived stigma of infertility. Therefore, legal and ethical changes are also required, in parallel with the social aspects, to ensure a smooth uptake. 

## 3. The Era of Big Data Is Enabling Better Prevention, Diagnosis and Treatments

Infertility can be attributed to men or women in equal proportions (20–30%), or to both partners (20%), with the remainder being unexplained. Female infertility is mainly associated with tubal damage, endometriosis, ovulatory dysfunction, or, rarely, uterine or cervical abnormalities. Broadly termed, male infertility based on origin could be divided into hormonal, testicular and post-testicular, affecting the quantity as well as quality of sperm [26]. A fundamental problem with developing personalised therapies or diagnostic tests for infertility is the limited understanding of the aetiology and pathophysiology of the disease. In fact, in about 10–25% of cases, the cause cannot be identified, leading to the so-called ‘unexplained infertility’ diagnosis. A refined diagnosis with a clear understanding of the mechanisms and underlying symptoms can lead to specific and effective clinical treatments, higher safety for patients, and reduced costs. A good example is the severe male-factor infertility due to the presence of varicocele. Surgical or radiological varicocelectomy can lead to improvements in semen parameters and an increase in chances of natural conception, or reduction in the need for surgical sperm retrieval [27]. Treatment of varicocele has also been associated with a reduction in the risk of pregnancy loss in couples with recurrent miscarriages (13.3% vs. 69.2%, *p* = 0.001) [28]. If left untreated, varicocele may lead to clinical hypogonadism with the associated negative effects of testosterone deficiency, poor sperm parameters or azoospermia [29]. Although varicocele is considered by some as a reversible cause of male infertility, these men are guided towards ART, ignoring the treatment that could restore normal testicular function and allow for natural conception [30]. 

The use of advanced next-generation sequencing (NGS) techniques—such as genomics, transcriptomics, proteomics and metabolomics—allows for the generation of a large volumes of biological and clinically useful data. These data can be used to support clinical decisions, uncover disease subtypes, associations, and prognostic or diagnostic biomarkers [31]. Figure 2 presents examples of the different types of data that are starting to be incorporated into an overall precision medicine picture in infertility. Several examples already exist in different areas of the reproductive sector, such as oocyte retrieval and selection [32], embryo culture and selection [33,34], embryo implantation prediction [35], male sperm selection [36,37], preeclampsia [38], and preterm birth [39].

### 3.1. Biomarkers and Screening Tests Can Guide Treatment Decisions

Biological markers (biomarkers), originally defined by Hulka and Wilcosky in 1988 [40], are essential tools and technologies in precision medicine. Biomarkers are biological entities that could be measured objectively and be used in the prediction, diagnosis and progression of a pathological process and used as indicators of responses to therapeutic interventions. 

One of the best examples of the use of biomarkers in reproductive health is the discovery of the role of angiogenesis-related factors, soluble fms-like tyrosine kinase-1 (sFlt-1) and placental growth factor (PlGF) in preeclampsia. sFlt-1/PlGF ratio through ML has been demonstrated as a valid biomarker for the identification of women at a high risk of preeclampsia and intrauterine growth restriction [41]. The increased production of autoantibodies, such as anti-phospholipid (APL) and/or anti-nuclear antibodies (ANA), were shown to be involved in infertility disorders including premature ovarian insufficiency (POI), unexplained infertility, as well as unsuccessful IVF treatments, preeclampsia, and spontaneous abortions [42,43]. For couples with recurrent miscarriages, the detection of serum APL and subsequent treatment with a combination of heparin and aspirin can lead to a reduction in pregnancy loss by 54% [44].

Biomarker discovery can be based on other types of information, such as non-coding RNAs, where some authors propose the use of non-coding RNAs and microRNAs as being predictive of male infertility [45,46], placental function during pregnancy [47], and of circulating small non-coding RNAs in the first trimester [48] as a non-invasive diagnostic tool for preeclampsia.

Possibly one of the most promising examples where ML was used to develop a predictive screening test to improve IVF success was the Endometrial Receptivity Analysis (ERA). The test assesses the expression of 238 genes that have been demonstrated as potential transcriptomic predictors of endometrial receptivity and allows for the personalisation of embryo transfer to allow for optimal synchrony between the embryo and endometrium [49,50]. The test still requires validation in a randomized controlled trial vs. conventional embryo transfer strategy to prove its clinical value, and there is still debate and conflicting evidence on its clinical benefit [51,52]. It is, therefore, important that the effect of any potential biomarker is replicated in a large independent cohort, with well-selected control populations before its adoption in the clinic.

Access to a diverse set of molecular profiling and other health-related data will not only be important for biomarker identification in isolation, but, when applied together in an integrated analysis, can improve the ability to form an accurate diagnosis. For instance, the serum cancer antigen 125 (CA-125) is the most extensively used peripheral biomarker in deep-infiltrating endometriosis (DIE) for detection of the disease and evaluation of therapy [53]. While studies that evaluate the performance of CA-125 have presented different limitations for the diagnosis of early stages of the disease, mainly in relation to their sensitivity [54,55,56], recent data-mining approaches modelling CA-125 levels in conjunction with other clinical features, such as transvaginal ultrasound, cyst or fallopian tube pathology, BMI and dyspareunia, have been shown to improve the sensitivity of this biomarker [57,58].

### 3.2. Mechanistic Understanding of Disease Can Help Stratify Patients for Treatment

Disease classification has always been challenging, especially in the context of biological heterogeneity, where the same symptom might be generated by different disease mechanisms. It should be possible to expand beyond broad distinctions of infertility symptoms, such as pelvic pain, dysmenorrhea, or metrorrhagia, and include molecular profiling and biomarker testing to classify diseases and disease subgroups. Genetic information, in the form of single-nucleotide polymorphisms (SNP), whole-exome sequencing (WES) or whole-genome sequencing (WGS), in combination with other omics data, are crucial in this context (Figure 2). One of the best examples of the use of genomic profiling in personalised medicine is Herceptin-targeted therapy, which is used to treat HER2-positive breast cancer [59]. In the infertility sector, endometriosis is a good example of the increased recognition of varied clinically informative molecular phenotypes, which can be used to segment patients into clusters of health relevancy (as reviewed in [60]). Initiatives such as the World Endometriosis Research Foundation (WERF) Endometriosis Phenome and Biobanking Harmonisation Project (EPHect) [61] are paving the way to advance biomarker and targeted treatment discovery in endometriosis.

Several recent reviews have described the complexity of female and male fertility [62,63,64], revealing an emerging picture of the high levels of genetic heterogeneity in certain diseases, such as PCOS, which can include variants in or near the Luteinizing hormone (LH), the FSH receptor (FSHR) genes or variants in the *FSH-β* gene [65]. A better understanding of disease mechanisms and related genes can guide the selection of drugs or treatment protocols, minimize harmful side effects, or ensure more successful outcomes [66]. For example, variants in the STK11 gene were associated with a decreased chance of ovulation in PCOS women treated with metformin [67]. Other examples include the identification of M2-carrier pregnancies by screening both partners for the M2 haplotype, which can be used to stratify couples for treatment with low-molecular-weight heparin (LMWH) [68].

### 3.3. Integrative Modelling of Non-Genetic Exposures Could Help Infertility Prevention Strategies

The exposome is a measure of the impact of exposure (e.g., diet, lifestyle) or an individual’s experience over their lives on their health [69]. It is well-established that certain environmental exposures and lifestyle factors may influence reproductive wellbeing; these include the consumption of coffee (WHO recommends <3 cups of coffee/day to minimise risk of miscarriage) [70], alcohol intake (moderate alcohol intake does not seem to be associated with infertility) [71], air pollution [72], and exposure to endocrine-disrupting chemicals (EDC), such as parabens, pesticides or BPA [73]. Several epidemiological and experimental studies have shown that exposure to EDCs can cause redox toxicity via oxidative stress and disrupt the hypothalamic–pituitary–gonadal (HPG) axis, thereby affecting male and female fertility [74,75]. For example, research has found that exposure parabens and pesticides is associated with reduced gestational age and increased risk of miscarriage [73]. Other studies observed a link between BPA and increased risk of miscarriage [76], and semen quality in men [74]. The relationship between EDCs and endometriosis is also being studied [77].

However, conducting exposome research can be challenging due to the complexity of study design, sample collection and data analysis, which must account for multiple confounders and variables. In fact, on some occasions, there have been reports of conflicting evidence [78]. In addition, it is widely recognized that there are interactions between an individual’s genetics and their sensitivity to environmental exposures [79]. This means that, in some cases, individuals with the same exposure levels may not develop the same disease. To better characterize the effect of multiple exposures and identify genetic interactions, blood and other body fluid samples would be required, along with genetic biomarkers in the same individual. Health policy and patient education will benefit from such studies [79]. Actions in this direction are visible in cancer research, for example, The Women Informed to Screen Depending on Measures of Risk (WISDOM) study evaluates the efficacy of risk prediction based on screening, clinical risk factors (e.g., BMI, age), breast density, and polygenic risk scores, to inform breast cancer screening [80,81]. More evidence, clearer and larger studies are needed and could be detrimental to the development of preventive strategies such as improving diet structure, eliminating exposure to certain toxicants, or even considering the use oocyte cryopreservation earlier for certain patients at a higher risk of infertility.

### 3.4. Genetic Data Can Be Used to Define Optimal Controlled Ovarian Hyperstimulation (COH) Dosing Regimens

While the standard approach to ovulation induction works for the majority of women undergoing IVF treatments, a small proportion will require more or less stimulating medication to achieve the same outcome. A proportion of women undergoing COH with gonadotrophins may underrespond, or not respond at all, to medications, while, in other cases, very high doses of gonadotrophins need to be used to achieve follicle growth despite objective good ovarian reserve. This is related to personal sensitivity to the given medication and the metabolism of receptor variants.

Clomiphene citrate (clomid) is used as a first-line treatment for PCOS. Nevertheless, variations in responses to clomiphene have been found, and cases of resistance have been reported. Studies have linked this event with variants in the enzyme cytochrome CYP2D6, as the drug is mainly metabolized into its active components by this enzyme [82]. Small studies have produced conflicting results [83,84]. Larger datasets may be a step towards personalized medicine in the field of ovulation induction for women with PCOS and to decrease the risk of cycle cancellation.

Mutations in the FSHR gene have also been found to contribute to this circumstance. Inactivating receptor mutations are known to occur in the FSHR gene [85,86], but there are other, more subtle alterations to the gene that modulate its activity. Single-nucleotide polymorphism (SNP) (G/A) in the 5′ untranslated region (UTR) of the FSHR gene at the −29 position is associated with altered transcriptional activity of the receptor gene [87], with the AA genotype being associated with poor response. In a small study of 50 women undergoing ART, one of the 7 women with AA genotype conceived (14%) vs. 31% of the GG type and 59% of the GA type. AA genotype women seem to require higher doses of gonadotrophins with lower number of antral follicles and a lower number of retrieved oocytes compared to the GG and GA genotypes [88]. When another SNP Asn680Ser was analysed, homozygous women for Ser680 achieved lower levels of oestradiol during COH [89], an effect that could be overcome by higher doses of gonadotrophins [90], with other studies refuting this result, using the effect of the studied population as a possible explanation [91,92,93]. Homozygosity for A in the Thr307Ala SNP in another small study has been associated with ovarian hyperstimulation syndrome (OHSS) (6 of 7 patients (86%), vs. 3 of 12 subjects with homozygous T allele (25%) and 6 of 31 (20%) subjects with heterozygous TA allele) [93].

It would, therefore, be advantageous and immensely helpful to know these facts before starting treatment in order to improve the outcome and reduce frustration due to a trial-and-error approach. Blood tests with an analysis of specific parts of the genome would be able to identify the small proportion of women with these variants and lead to improved treatment results, minimising time to pregnancy and associated costs. To make this mainstream, larger studies on a more diverse population are required; alternatively, ML/AI could be employed to assess the genetic diversity that may have eluded normal analysis.

### 3.5. The Microbiome as an Important Emerging Health Data Stream in Infertility

The female reproductive tract microbiota accounts for 9% of the bacterial load in humans, with *Lactobacilli* being the dominant genus in a healthy woman [94]. Bacteria have been isolated from every part of the female reproductive tract, including the peritoneal fluid of the pouch of Douglas, where the concentration of bacteria was 10,000 times lower compared to that of vaginal fluid [95]. Different bacteria have also been isolated from the seminal fluid of healthy and infertile men, highlighting that *Lactobacillus* also plays a protective role in sperm health [96,97]. NGS has allowed to characterise in detail the microbiota of the female reproductive tract and has shown that the percentage composition of *Lactobacillus* in the endometrium differs between healthy volunteers (85.7%), non-IVF patients (73.9%), and IVF patients (38%) indicating that the majority of infertile patients demonstrate an abnormal endometrial bacterial profile [98,99]. The domination of *Lactobacillus* in the endometrial microbiota (96.5% ± 33.6%) in the same infertile population was associated with pregnancy, suggesting that *Lactobacillus*-dominated bacterial flora might favour implantation [98]. The isolation of pathogenic bacteria or diagnosis of chronic endometritis, a condition associated with recurrent implantation failure, may improve the chance of healthy-term pregnancy with a course of antibiotics before embryo transfer [100,101]. The use of probiotics containing the *Lactobacillus* genera could be an alternative to avoid the necessity for antibiotic treatment and its side-effects; however, evidence to support this intervention is lacking. With mounting evidence for the importance of reproductive tract microbiota for reproductive health, more pressure should be put into assessment of this before treatment, especially in the population of couples with recurrent implantation failures or recurrent miscarriages. The data obtained during such tests could add to the aspect of personalized medicine and, with the help of ML/AI models, an exact microbiome-print could be identified, which assists the conception and continuation of pregnancy to term.

## 4. Machine Learning Is Aiding ART Treatments

Since Louise Brown, the first IVF baby, was born in 1978, it is now estimated that over eight million babies worldwide have been conceived through assisted reproduction. The overall success rates of achieving a pregnancy in couples undergoing ART are marginally better than the chances of natural conception. Whilst we are proficient in obtaining oocytes and spermatozoa and creating embryos, replacing them in the uterus, we have almost no influence and limited knowledge of the events that occur thereafter. Regarding research funding in obstetrics and gynaecology is concerned, fertility research is perceived as an ugly duckling compared with funding for gynaecological oncology. In 2018/19, Cancer Research UK, spend £546 M on research [102], whereas NIHR, in the preceding year, spent only £21M in research grants across reproductive health and childbirth combined [103]. The COVID-19 pandemic has most likely had a further detrimental effect on the levels of funding available for this type of research. This disparity likely originates in the fact that IVF centres are perceived as commercial entities, with research not being a priority. This low-level funding for research in the field of ART is concerning when combined with the fact that, in the last decade, a decline was observed in worldwide IVF birth rates. This is likely to be attributed to mild stimulation protocols, pre-implantation genetic testing for aneuploidy (PGT-A), freeze-all cycles, embryo banking, increased industrialization, and the unchecked introduction of IVF add-ons, with very little good-quality research being conducted to assess the actual benefit of any of these interventions on pregnancy outcomes [104,105,106].

In recent years, new technologies have emerged with the purpose of increasing live birth rates after IVF. For instance, time-lapse imaging allows for the monitoring of the development of the embryos and, as such, the selection of the most promising one, without relying only on the subjective assessment of morphology by the embryologist [33,34]. Non-invasive assessment of the embryo culture fluid can, with some degree of accuracy, be used to determine the ploidy status of the embryo without resorting to invasive biopsy [107]. Recent developments looking at the endometrial receptivity [50,108] to personalize embryo transfer and align it with the individual window of implantation have shed some light on the workings of the endometrium. All these advances were carried out to create more objective ways of selecting the best embryos and improving the chances of embryo implantation. However, many suggested ‘add-ons’ to fertility treatments seem to be carried out without guidance and selection and, in many cases, there is insufficient evidence of their added benefits.

There are many parameters and factors contributing to the chance of a successful ART treatment. Without accurate and quantitative data, an overarching approach is not possible, other than the clinicians’ experience in a somehow subjective risk/chance calculation. ML/AI could incorporate all the minute details that are often overlooked and provide superior guidance for the fertility specialist. 

The growing role of ML/AI in enhancing embryo selection and evaluation, and ultimately improving pregnancy rates, has brought massive enthusiasm and investment. The adoption of AI-based methods has significantly increased in IVF clinics in recent years [109]. For the interested reader, there are recent reviews discussing examples in depth in the literature [109,110]. In this section, we look at a few examples and use cases on how ML/AI is transforming ARTs (Figure 3).

Integrating patient data analysis as a first step in ART treatment is now starting to be considered in IVF treatments, with a few recent examples of studies demonstrating improvements in the recombinant follicle-stimulating hormone (rFSH) dosing using algorithms based on patient’s data, such as BMI and AMH concentration [111,112]. Other baseline characteristics entered into AI algorithms would allow for the appropriate counselling of the couple regarding their chance of success per treatment cycle started, if they require any additional treatments to improve their fertility potential, how many cycles they may require and allow for financial planning around this treatment. In cases where the chance of success is very low (women with very poor ovarian reserve or significant comorbidities), with the help of these algorithms, the clinician could accurately counsel the couple regarding the best course of action, which, in these cases, may be oocyte donation, surrogacy or adoption. Time to pregnancy in such cases would be shortened and the significant financial impact on the couple would be mitigated.

Machine learning and AI also create huge opportunities for improving the selection of spermatozoon for injection in an ICSI procedure. At present, sperm selection for ICSI is subjective, based on assessments made by clinical experts [113]. The research and development of computer-based methods that can provide objective assessments of sperm kinematics, sperm morphology and DNA integrity has been booming [113,114]. This research is extremely important to enable the development of next-generation ML models in sperm selection with the hope of maximising the chance of pregnancy in the future.

Oocyte quality assessment is another area in which ML/AI is making significant progress. AI can be applied to analysing oocyte images to identify oocytes with the best reproductive potential (competent oocytes), something that a trained embryologist cannot do at present [110]. AI methods could not only improve the ART performance, but also provide an accurate evaluation of oocyte vitrification and warming procedures [115], which is of particular interest, as oocyte cryopreservation is becoming increasingly popular. Additionally, by using non-invasive AI methods, the quality of the ovarian yield could be analysed retrospectively after an ovarian stimulation cycle, which may be helpful in managing psychosocial expectations of intended parents.

The analysis of ultrasound images of the endometrium in combination with any molecular receptivity data in a combined algorithm, could provide a more accurate assessment of endometrial receptivity than either of these techniques used alone. Such an approach is not yet being utilized in the fertility clinic. Following oocyte collection, computer vision and AI that uses advanced machine learning algorithms to understand static images of a day 5 blastocyst are already being utilized to predict embryo viability and chance of pregnancy with a 24.7% improvement in accuracy compared to a trained embryologist [116]. This model is used commercially by fertility clinics worldwide. Static image analysis and AI algorithms have similarly been employed in assessing the risk of embryo aneuploidy and have reached concordance level with the biopsy result of 81.5% [117]. 

Once the pregnancy has started, there is very little that can be offered to the couple in terms of estimation of a successful outcome. Depending on the healthcare system, a pregnant woman contacts her gynaecologist or general practitioner and either has a blood test or a scan. A doubling of hCG levels every 48 h has been used for a long time as a predictor of the chance of a viable intrauterine pregnancy vs. a risk of an ectopic pregnancy [118]. This approach is mainly used in case of early pregnancy complications, such as pain and or bleeding and rarely employed in low-risk pregnancies, other than maybe IVF pregnancies. Recently, ML has shown some promise in providing some useful guidance on follow-up frequency for patients thought to be at high risk of miscarriage after IVF [119]. Ultrasound imaging is the mainstay of pregnancy assessment; however, its accuracy is largely dependent on the operator. The advent of 3D ultrasound and the ability to store a volume of the scanned area allows for the storage of a dynamic image of an entire organ and offline or even off-site analysis by different operators. It would, therefore, be possible to employ AI to combine biochemical markers of pregnancy with 2D or 3D images of the early pregnancy to provide the couple with risk of miscarriage, preterm delivery and ultimately, live term pregnancy. The utilization of algorithms in early pregnancy has been used for trisomy screening, where nuchal fold thickness, in combination with free human chorionic gonadotrophin beta (beta-hCG) and pregnancy-associated plasma protein A (PAPP-A), are used to calculate a risk of trisomy 21 [120]. This approach is likely to be superseded by NIPT- non-invasive prenatal testing of free fetal DNA from maternal blood, which can be performed as early as 7 weeks gestation. This is far more reliable than the combined test for diagnosis of trisomy 21, 18 and 13, but has a higher overall cost [121]. Soft ultrasound markers of chromosomal anomalies are utilized by the clinicians as a means to carry out more definitive tests, such as chorionic villous sampling or amniocentesis, both of which are associated with a 1–2% risk of miscarriage [122]. This may be unacceptable to some couples, and they would welcome a non-invasive assessment combining what is already known about the pregnancy and providing them with a possible outcome. AI could incorporate all this particulate knowledge into one chance-of-healthy-term pregnancy algorithm. This would have to be a dynamic process as the pregnancy develops and more items can be inputted. The patient having access to these data, and the ability to input new data, would make it a valuable tool for the couple, guiding them as to the need for medical attention. 

Finally, mobile applications, such as the QUIPP app [123] utilize the clinical data regarding patients’ previous pregnancies, their duration, any cervical surgery, cervical length measurement and levels of fetal fibronectin in the vaginal secretions to predict the risk of preterm delivery with an area under the curve of 0.763 for delivery <34 weeks and 0.746 for delivery <37 weeks when the cervical length and fetal fibronectin were entered between 22 + 0 and 25 + 6 weeks gestation [124]. 

## 5. The Participatory Citizen: From a Disease-Centric Model to Active Wellness

The term ‘Participatory medicine’ was famously defined by the patient activist Giles Frydman in 2010 as ‘a movement in which networked patients shift from being mere passengers to responsible drivers of their health, and in which providers encourage and value them as full partners’ [125]. The term is one of many that are used interchangeably, including ‘patient-centric medicine’, Health 2.0, Medicine 2.0, e-medicine, and eHealth. Among the various causes of infertility, some are related to factors that can be modulated by individuals themselves, such as an unhealthy lifestyle and diet. In addition, lack of awareness and ‘fertility education’ may hinder the search for treatments. The basic idea is that an active and engaged individual can not only become an active player, influencing their own health and well-being, but also a good advocate of preventative and early detection strategies. Participatory health efforts are largely intertwined with the availability of information and relevant data, and include the use of mobile health apps (often referred to mHealth), self-tracking records, social health networks and direct to consumer (D2C) tests such as genomics and blood tests (Figure 1). 

In recent years, we have seen an explosion of apps, wearables, and tools powered by ML dedicated to optimizing the chance of pregnancy and reproductive wellbeing. The digital market and support resources for women seeking fertility services is skyrocketing. Period-tracking apps, which aim to help with cycle-monitoring and pinpointing ovulation, such as Glow and Clue, have millions of users. Self-tracking apps are not only winners for female reproduction but for anything trackable, such as steps, calories, diet, sleep patterns, mood, and physiological parameters such as blood pressure, heart rate, and pulse rate. Various Machine Learning techniques are often applied to recommender systems, helping to instigate behavioural changes in the user. ML/AI is undoubtedly unlocking the value of self-tracking data, increasing awareness and supporting health behaviours; however, weather these changes have a real impact on patient’s fertility treatments or pregnancy still needs to be demonstrated and validated.

A large proportion of female digital healthcare companies focus on making health-related tests available directly to consumers. These can range from at-home blood tests for AMH and other hormones (e.g., Parla and Modern Family), to turning a smartphone into a powerful microscope to allow for users to test their semen at home (e.g., YO test and ExSeed health). In 2017, Celmatix Inc, a USA-based company, announced the first commercially available product that provides D2C genetic testing, the Fertilome. The Celmatix test currently examines 49 SNPs in 32 genes, which have been implicated in a variety of reproductive conditions, such as POI, recurrent pregnancy loss (RPL) and PCOS. This genomic test was one of the first ones to offer significant amounts of genomic data related to reproductive health directly to individuals, without the mediation of medical professionals. Insights from this test could help to identify infertility- (such as diminished ovarian reserve) and pregnancy-related complications earlier. However, this level of access to genomic information has been criticized [126,127] as it may cause unnecessary anxiety for patients and clinicians, especially when it comes to explaining the implications of problematic results that might not be relevant to the patient. For example, in some cases, there might be inadequate evidence (e.g., lack of appropriate controls, underpowered studies, ethnic specificity) to accurately interpret the pathogenicity of a certain genetic polymorphism. Additionally, due to the incomplete understanding of the processes related to human fertility, the results of such tests may be irrelevant until the woman tries to conceive. The implications of such tests may vary; for example, finding out you are a carrier of a FSHR mutation, reducing its sensitivity to COH, which may or may not be needed in the future, is of lesser significance than finding out you are a carrier of the BRACA-1 gene mutation, giving the affected person an up to 87% lifetime risk of developing breast cancer and up to 63% risk of developing ovarian cancer [128]. Furthermore, we still have a very limited understanding of genetic risk. Probabilistic risk information for any health condition remains difficult to be actionable in the clinic, such as advising patients to make behavioural changes or guide clinicians into choosing risk-reducing interventions. In the future, ML/AI algorithms could aid in the process of assigning significance to these results and making them a valuable tool in e-medicine.

The COVID-19 pandemic has dramatically changed the story of telehealth. Patients and clinicians are now more receptive to virtual care. We are living in unprecedent times when it comes to embracing D2C “at-home” testing, health mobile apps and self-monitoring and tracking tools; thus, we can trust that these technologies will continue to gain momentum in the future. eHealth may prove extremely helpful, enabling physicians to reach more patients and providing valuable real-world data (RWD) to inform research, treatment development and patient care, and giving individuals more control of their reproductive health. The potential is big; however, the full capabilities and inevitable shortcomings must be properly validated before this potential value can be realized [129].

## 6. Challenges to the Use Machine Learning and Big Data in the Infertility Sector

Despite its numerous high-value benefits, ML and AI are not without challenges (Figure 4). Key challenges include (1) difficulties in obtaining accurate and high-quality data, (2) the challenges of generalisation and unintended algorithmic and dataset biases, (3) validation of ML/AI algorithms and (4) translation into clinical practice. Developers of AI models must be cognizant of these limitations to capture the maximum amount of value from the predictive models.

### 6.1. High Quality and Quantities of Data

The first step to implementing an ML/AI algorithm is access to Big Data, as many clinics have not yet implemented rich electronic health record (EHR) systems. The UK Human Fertilisation and Embryology Authority (HFEA) prospectively collects baseline information and birth outcomes on all licensed fertility treatment cycles performed in the UK since 1991. It is one of the biggest data collections publicly available in the world, comprising more than 1.3 million treatment cycles up to 2018. However, many important predictors such as BMI, detailed clinical history, and lifestyle parameters, and many details regarding the sperm, oocyte, and embryo quality assessments, are not robustly collected in this dataset, limiting its use for modern ML applications. For an ML model to be consistently successful at making predictions, data must be of high quality and of sufficient sample size; the quality of the data is as crucial as the quantity of data. If data are not labelled or categorized appropriately, or not in sufficient amounts to represent an unbiased sample of the general population, then the model will be unduly influenced by noise in the data. The number of records required to reach sufficiently high model performance depends on the model itself, and the number of predictor variables that are required. The rule of thumb for a regression model is to have 20 observations for each parameter [130], however this number will vary if the task at hand is more complex, such as an image classification task using deep learning. Although deep learning requires larger amounts of data than traditional ML, the use of more data does not always impact the performance. Using well-understood, strong predictor variables can decisively affect model performance. A recent study on type 1 and type 2 diabetes has demonstrated that traditional logistic regression models can perform as well as complex machine learning algorithms such as neural networks and gradient boosting machines, provided that the data contain a set of strong predictor variables [131]. Similarly to this example, models of oocyte quality assessment could benefit from combining a typical ‘black-box’ approach (in which we let the computer identify the general features that are important for classification), with a more targeted approach where additional predictor variables are chosen based on domain knowledge. These added features should be related to well-known factors that affect oocyte quality, such as patient age, BMI, and smoking status. In summary, the data used for training a ML algorithm should not only be of high-quality, but also contain the necessary predictor variables and be representative of the problem domain. Finally, and equally importantly, data should be compliant with the ‘FAIR’ data principles, meaning that they are Findable, Accessible, Interoperable, and Reusable [132]. The public availability of patient data has been endlessly discussed. While it is important to restrict access to sensitive patient information, we need to improve and facilitate data access requests in the reproductive health field (in many cases, an access request is made by emailing the corresponding author of a publication). Making research datasets easily available to the wider community will undoubtedly help to accelerate the research progress.

### 6.2. Generalizability of Learning

A generalizable model is a model that is not only representative of the population used for training but also transposable to other populations. For instance, it is well recognised that infertility investigations and treatment practices tend to vary across clinics (due to differences in population demographics or socio-economic statuses, differences in treatment regimens, measuring devices, laboratory protocols, or compliance with established guidelines), countries (e.g., due to differences in population or IVF regulations) and temporal periods (e.g., due to advances in technology). Thus, to build generalizable models, it is important to collect data across different clinical settings, populations, and subgroups of interest. If the data are small and based on local clinics or centres, they may not represent the real-word setting because there might be clinic- or demographic-specific nuances in certain clinics. In fact, the treatment centre is a well-known, very strong predictor of the success of IVF [133], as many indications for IVF treatment are defined qualitatively, and their use might vary among physicians of different clinics. On the other hand, if the sample size is large, the data should have a wide variety of records, including unique or odd cases, to produce a generalizable model. If not, the models derived using multi-centre national databases could have poorer quality when compared to single-centre-collected and physician-curated data. Hence, it is important to check that the population and data collection are in accordance with, and representative of, the envisioned application scenario, and meaningful to patients in the context of their clinics. 

#### 6.2.1. Data Biases Introduced by Population Heterogeneity 

There are several examples of heterogeneous model performance across different settings and populations. This is often defined as a ‘spectrum effect’ or ‘spectrum bias’, a term used to describe the situation where test performance varies across patient populations [134]. Failure to recognize and address heterogeneity will lead to models that are not generalizable [134,135]. One key source of heterogeneity is ethnical diversity. For example, it has been shown that ethnicity plays a role in IVF success rates [136]. Another important, and often missed, cause of heterogeneity is the prevalence of the different treatment types performed in different countries, or even in different clinics of the same country. For instance, the usage rates of ICSI vs. standard IVF vary from region to region, with a 55% usage in Asia, 65% in Europe, 73% in North America, and 86% in South America. The highest proportion of ICSI is utilized in the Middle East with almost all cycles using ICIS to fertilize oocytes [9,137]. Many other clinical and laboratory differences exist, such as types of COH regimes, protocols for oocyte retrieval, including sedation equipment setup; differences in treatment strategies, clinical guidelines, methodological standards, and experience; differences in disease definitions, etc. All these problems may lead to heterogeneity in the magnitude of the predictor effects [138], in the prevalence of the measured outcome, and in the distribution of predictor values themselves [139].

#### 6.2.2. Non-Stationarity in Treatment Data and Historical Biases

Historical biases can be engrained at any category of infertility-related data, but are particularly relevant when developing models around IVF treatment data. As the regulatory environment around IVF practices continues to change and treatment practices continue to evolve, great care must be taken when choosing which data to use and which to exclude. For instance, elective single-embryo transfer (eSET) is now a common practice across clinics in UK since the introduction of a policy to encourage fertility centres to eSET in 2009 [140,141]; the number of same-sex couples seeking IVF increased substantially after 2008 when the legal requirement for a father figure was removed (between 2014 and 2013; IVF increased by 20.1% for same-sex female couples, and by 2.7% when considering all couples) [142]; egg freezing increased by 10% from 2016 to 2017; egg thawing was used in 581 cycles in 2017 (compared to 159 in 2012) [142]; the number of frozen embryo replacements (FERs) increased by 86% between 2014 and 2019 [142]. These changes and evolving trends require a close partnership with ML experts and clinics to continuously improve the predictions models through the so-called ‘model updating’ [143], where continuous adjustments to the prediction models are performed and re-calibrated in new populations from time to time, to provide accurate estimates of success-rate changes over time.

### 6.3. Algorithm Validation Using Double-Blinded Datasets

Evaluating model performance across populations, clinics and countries is extremely important. A model performance can be quantifiable using metrics such as positive-predictive value, sensitivity, and specificity. Probability-prediction models can be evaluated using discrimination (AUC or C statistic) and calibration metrics. The performance of a machine learning model can be assessed using data from the same source as the training sample (e.g., 80:20 data split for train and test data). That is, 20% of the cases are removed before the data were modelled; these removed cases were called the testing set. Once the model was built using the 80% of remaining cases (often called the training set), the cases that were removed (testing set) can be used to test the performance of the model on the “unseen” data (i.e., the testing set). However, a true evaluation of generalisability (also called transportability) typically requires the external validation of a prediction model [144]. This external evaluation should be done in a completely blinded way; in some cases, this is named ‘double blinded’ evaluation [116], which consists of using data that not only have never been used in the training of the algorithm but originated from a completely different and independent clinical environments, and, if possible, different countries. 

### 6.4. The Challenge of Translation to Clinical Practice

Challenges of translation of AI systems in infertility care include those inherent to the science of machine learning itself, such as difficulties in implementation and scalability in the deployment, but also the human barriers to AI adoption [145]. The human barriers to the clinical adoption of an AI algorithm depend on two main factors: clinical usefulness and trustworthiness [146]. Before implementation in the clinic, we need to clearly understand how AI models will affect the quality of care, how they will improve on the efficiency and productivity of clinical practice, and most importantly, how will they improve on patient outcomes. It is vital to demonstrate the added value of an AI algorithm when compared to legacy or conventional methods. Randomised controlled trials are viewed as the gold standard for robust clinical evaluation but conducting these in practice may not always be appropriate or feasible. If the algorithm is used to aid clinical decisions related to interventions that are expensive, invasive or have unwanted side-effects, we should be concerned with false-positive predictions that could result in unnecessary harm to the patient [121,146]. One example is the use of computer-aided diagnosis for mammography in the late 1990s, which, due to false-positive predictions, was found to significantly increase recall rate and surgical interventions without improving patient outcomes [147]. On the other hand, if the interventions are predominantly preventative or “assistive” (i.e., aim to provide additional assistance to patients), we would want to minimise in the number of false negatives [121,146]. To this end, thoughtful post-market surveillance should be implemented to make sure that the algorithm does not create or exacerbate inequalities [146]; for instance, if certain groups of patients are deprived of access to beneficial innovations (e.g., in minority ethnic groups, or certain age groups [148]). This so-called “unintended discriminatory bias” could occur when the algorithms are not generalizable to new populations. To overcome these potential pitfalls in the clinical adoption of any ML algorithm, it is important to systematically test the algorithms for bias and fairness [149,150].

## 7. Conclusions

Advanced algorithms associated with ML or AI have found their way into modern medicine and are transforming many aspects of patient care. Infertility and reproductive health are catching up [151]; however, much more can be done to improve the chance of conception for infertile couples. The major limitations include access to enough good-quality data to produce algorithms that are reliable and reproducible on a global scale. The social understanding and knowledge of reproductive ageing is increasing and, as such, a shift can be seen towards individuals being more engaged in their own reproductive future. This translates to increased uptake of fertility MOTs and social egg-freezing. As male reproductive ageing is not as pronounced, social sperm-freezing is not a common practice. We live in an exciting time of telemedicine, with ML/AI influencing every aspect of our lives. It is not unexpected that attempts at the integration of clinical experience and information technology into an amalgam aiming to improve health and prolong lifespan, as well as affect our reproductive health, are emerging everywhere. It is, however, important to critically appraise these and not blindly jump at any opportunity labelled ‘AI will increase your chance of pregnancy’. One must bear in mind that IVF is also a business and, as such, is governed by laws of the open market, where any new developments are firstly there to lure customers, often with no or limited regard for the service user. 

To improve the development and uptake of AI algorithms in reproductive medicine, major obstacles would need to be overcome. Our understanding of the processes governing reproduction are not fully understood and, as such, the data that will form the basis of such algorithms will be incomplete or, on occasion, incorrect. AI in the field or reproductive medicine may also be useful to sift through the plethora of the available scientific data and help identify aspects of further research that could be worth pursuing. With the expanding understanding of physiology, we would be able to identify the pathology and devise the individualized treatment protocols that would best suit the woman and guarantee reproductive success. Easy access to self-entered data via various apps may be of help in the former, but clinical input and experience in combination with data analysts and biostatisticians is necessary to increase the influence of ML/AI in reproductive health and move reproductive medicine into version 2.0.

## Figures and Tables

**Figure 1 jcm-11-06426-f001:**
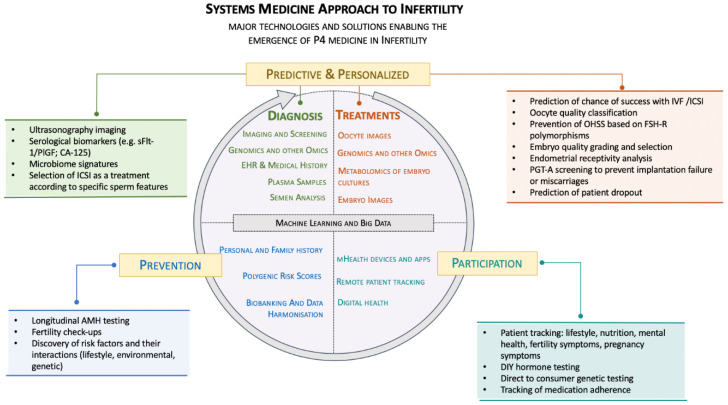
The systems approach to infertility. The figure shows the major technologies and solutions enabling the emergence of a P4 system in infertility. ‘Prevention’ refers to the ability to prevent infertility disorders or pregnancy complications. ‘Predictive and Personalized’ refer to the prediction and management of disease with greater granularity and the individualization of treatments. ‘Participation’ refers to patient participation in their own health. The text inside the circle indicates the key data sources generated by each segment, which can be integrated using ML and big data mining techniques. Text boxes outside the circles include practical examples of the use of these data sources. At the centre of the circle, we highlight ‘Machine Learning and Big Data’, which are essential to every of the four sections to power the innovation cycle of a ‘systems medicine approach’ to infertility.

**Figure 2 jcm-11-06426-f002:**
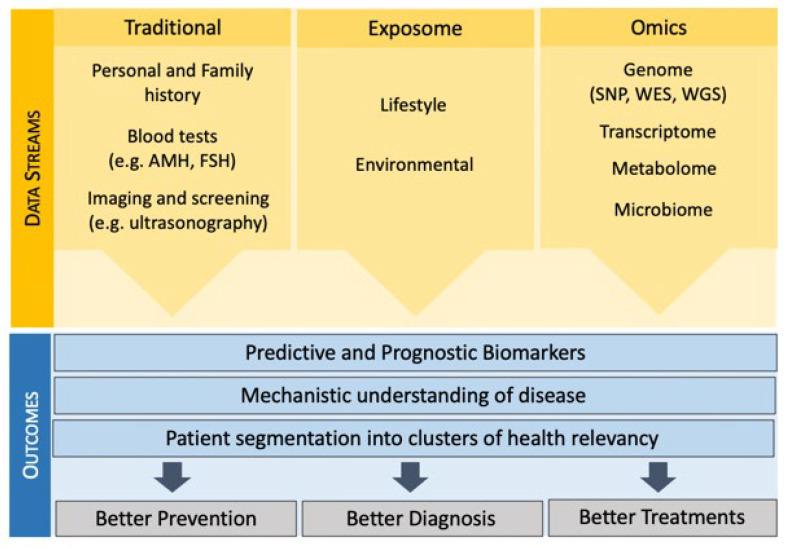
Data streams used to enable precision medicine in the Infertility sector. The newer health data streams such as genomics, the microbiome and the exposome are a complement to traditional approaches used in infertility investigations and management, to ultimately deliver better prevention, diagnosis and treatments.

**Figure 3 jcm-11-06426-f003:**
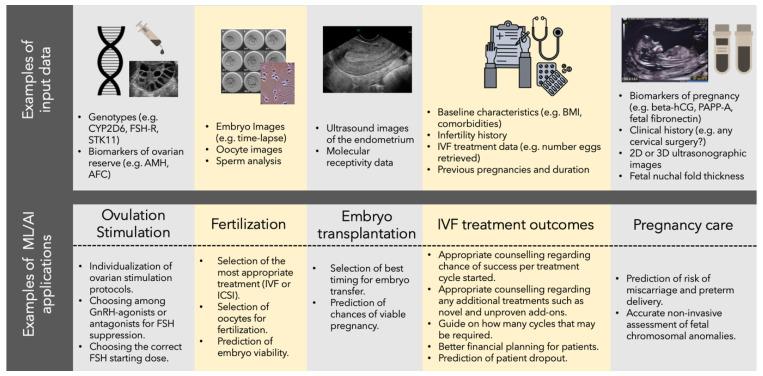
Examples of different types of input data and ML models that can be built and applied to improving ART treatments.

**Figure 4 jcm-11-06426-f004:**
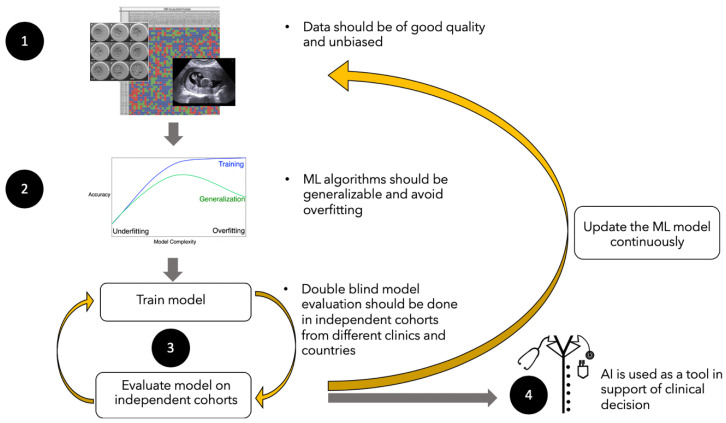
Challenges associated to the use of ML/AI algorithms. (1) Data should be labelled accurately and unbiased; (2) ML/AI algorithms should be generalisable and avoid overfitting; (3) Algorithms should be validated in independent cohorts and continuously updated; (4) Algorithms need to be translated and adopted into clinical practice.

## Data Availability

Not applicable.

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
