# Peer review of "Data-Driven Medicine in the Diagnosis and Treatment of Infertility"

_jcm, 2022, doi:10.3390/jcm11216426_

Round 1
Reviewer 1 Report
After evaluating the valuable study "Data-Driven Medicine in the Diagnosis and Treatment of 2 Infertility" I considered the following comment necessary to improve the quality of the article.
It would be better to add to the content about the effect of oxidative stress and the hypothalamus-pituitary-gonadal axis (help from the reference below).
Akbaribazm, M., Goodarzi, N., & Rahimi, M. (2021). Female infertility and herbal medicine: An overview of the new findings. Food science & nutrition, 9(10), 5869-5882.
Author Response
Dear Reviewer 1,
We appreciate the time and effort that you dedicated to providing feedback on our manuscript and are grateful for the insightful comments on and valuable improvements to our paper. We have incorporated most of the suggestions made by the reviewers. Those changes are highlighted within the manuscript. Please see below, in blue, for a point-by-point response to the reviewers’
comments and concerns.
Academic Editor Comments
Suggested article (pre_decision_comments-2.pdf)
Author response: We included a citation to the paper mentioned “Artificial intelligence in human in vitro fertilization and embryology (2020)” and a better introduction to this section (see lines 583-592)
Reviewer 1
It would be better to add to the content about the effect of oxidative stress and the hypothalamus-pituitary-gonadal axis (help from the reference below). (https://pubmed.ncbi.nlm.nih.gov/34646552/)
Author response: Thank you, we have included a more detailed discussion related to oxidative stress, the exposome and infertility in the article now (see lines 372-443)
Reviewer 2
- Please check the consistency of the font in Figure 1.
Author response: Thank you, we have checked all the fonts in the figure. We did not find inconsistent fonts.
- The reviewer suggests the authors briefly summarized and shorten the content in Page 3 Lines 87-95.
Author response: Thank you, changes done.
- The role of ML/AI and the association between data-driven medicine and the conceptual shift from a disease-centric to a health-centric model should be described more clearly and highlighted.
Author response: Thank you, changes done.
- In section 2 "The era of big data is enabling better prevention, diagnosis and treatments", subheadings were encouraged to increase the readability.
Author response: Thank you, changes done.
- In Page 8 Lines 339-374, the content of genomic data did not mention the machine learning method therefore should be described in section 2.
Author response: Thank you, changes done.
- Similar to question 3, please describe more about the role of ML/AI and the association between data-driven medicine and the "participatory citizen" in section 4.
Author response: This section is how the patient is using data and apps to guide their decisions about their own health. Here, AI/ML are having an impact but the section focuses more on the availability of apps, telehealth and Direct-to-consumer testing. We re-phrased some of the sentences.
- The reviewer recommends the use of "challenges" instead of "limitations" in the heading of section 5.
Author response: Thank you, changes done in section 5 and in the abstract
Reviewer 2 Report
In this opinion article, the authors discuss recent influential work on the application of big data and AI in the prevention, diagnosis and treatment of infertility. They evaluated the limitations of the sector and present an interpretation of the different innovation forces that are driving the emergence of a systems approach to infertility care. Efforts including the integration of multi-omics information, collecting well-curated biological samples in specialised biobanks, and stimulating the active participation of patients were considered, which is of great significance for the clinical diagnosis and treatment of it. In general, the manuscript has collected a relatively rich literature, also in place, and furthermore, it has a good significance for summary.
1. Please check the consistency of the font in Figure 1.
2. The reviewer suggests the authors briefly summarized and shorten the content in Page 3 Lines 87-102.
3. The role of ML/AI and the association between data-driven medicine and the conceptual shift from a disease-centric to a health-centric model should be described more clearly and highlighted.
4. In section 2 "The era of big data is enabling better prevention, diagnosis and treatments", subheadings were encouraged to increase the readability.
5. In Page 8 Lines 339-374, the content of genomic data did not mention the machine learning method therefore should be described in section 2.
6. Similar to question 3, please describe more about the role of ML/AI and the association between data-driven medicine and the "participatory citizen" in section 4.
7. The reviewer recommends the use of "challenges" instead of "limitations" in the heading of section 5.
Author Response
Dear Reviewer 2,
We appreciate the time and effort that you dedicated to providing feedback on our manuscript and are grateful for the insightful comments on and valuable improvements to our paper. We have incorporated most of the suggestions made by the reviewers. Those changes are highlighted within the manuscript. Please see below, in blue, for a point-by-point response to the reviewers’
comments and concerns.
Academic Editor Comments
Suggested article (pre_decision_comments-2.pdf)
Author response: We included a citation to the paper mentioned “Artificial intelligence in human in vitro fertilization and embryology (2020)” and a better introduction to this section (see lines 583-592)
Reviewer 1
It would be better to add to the content about the effect of oxidative stress and the hypothalamus-pituitary-gonadal axis (help from the reference below). (https://pubmed.ncbi.nlm.nih.gov/34646552/)
Author response: Thank you, we have included a more detailed discussion related to oxidative stress, the exposome and infertility in the article now (see lines 372-443)
Reviewer 2
- Please check the consistency of the font in Figure 1.
Author response: Thank you, we have checked all the fonts in the figure. We did not find inconsistent fonts.
- The reviewer suggests the authors briefly summarized and shorten the content in Page 3 Lines 87-95.
Author response: Thank you, changes done.
- The role of ML/AI and the association between data-driven medicine and the conceptual shift from a disease-centric to a health-centric model should be described more clearly and highlighted.
Author response: Thank you, changes done.
- In section 2 "The era of big data is enabling better prevention, diagnosis and treatments", subheadings were encouraged to increase the readability.
Author response: Thank you, changes done.
- In Page 8 Lines 339-374, the content of genomic data did not mention the machine learning method therefore should be described in section 2.
Author response: Thank you, changes done.
- Similar to question 3, please describe more about the role of ML/AI and the association between data-driven medicine and the "participatory citizen" in section 4.
Author response: This section is how the patient is using data and information to guide their decisions about their own health. Here, AI/ML are having an impact but the section focuses more on the availability of apps, telehealth and Direct-to-consumer testing. We re-phrased some of the sentences.
- The reviewer recommends the use of "challenges" instead of "limitations" in the heading of section 5.
Author response: Thank you, changes done in section 5 and in the abstract